# Factors Associated with Acute Community-Acquired Pyelonephritis Caused by Extended-Spectrum β-Lactamase-Producing *Escherichia coli*

**DOI:** 10.3390/jcm10215192

**Published:** 2021-11-07

**Authors:** Mónica Romero Nieto, Sara Maestre Verdú, Vicente Gil, Carlos Pérez Barba, Jose Antonio Quesada Rico, Reyes Pascual Pérez

**Affiliations:** 1Department of Internal Medicine, Elda General University Hospital, Elda/Sax, Road s/n, 03600 Elda, Alicante, Spain; saramaver@gmail.com (S.M.V.); cperezb@coma.es (C.P.B.); 2Department of Clinical Medicine, Elche Miguel Hernández University, 03550 San Juan, Alicante, Spain; vte.gil@gmail.com (V.G.); jquesada@umh.es (J.A.Q.R.); 3Research Unit, Elda General University Hospital, 03600 Elda, Alicante, Spain

**Keywords:** acute pyelonephritis, *Escherichia coli*, extended-spectrum ß-lactamase

## Abstract

This study aimed to identify the factors associated with the presence of extended-spectrum ß-lactamase-(ESBL) in patients with acute community-acquired pyelonephritis (APN) caused by *Escherechia coli* (*E. coli)*, with a view of optimising empirical antibiotic therapy in this context. We performed a retrospective analysis of patients with community-acquired APN and confirmed *E. coli* infection, collecting data related to demographic characteristics, comorbidities, and treatment. The associations of these factors with the presence of ESBL were quantified by fitting multivariate logistic models. Goodness-of-fit and predictive performance were measured using the ROC curve. We included 367 patients of which 51 presented with ESBL, of whom 90.1% had uncomplicated APN, 56.1% were women aged ≤55 years, 33.5% had at least one mild comorbidity, and 12% had recently taken antibiotics. The prevalence of ESBL-producing *E. coli* was 13%. In the multivariate analysis, the factors independently associated with ESBL were male sex (OR 2.296; 95% CI 1.043–5.055), smoking (OR 4.846, 95% CI 2.376–9.882), hypertension (OR 3.342, 95% CI 1.423–7.852), urinary incontinence (OR 2.291, 95% CI 0.689–7.618) and recurrent urinary tract infections (OR 4.673, 95% CI 2.271–9.614). The area under the ROC curve was 0.802 (IC 95% 0.7307–0.8736), meaning our model can correctly classify an individual with ESBL-producing *E. coli* infection in 80.2% of cases.

## 1. Introduction

Acute pyelonephritis (APN) accounts for a large proportion of community-acquired and hospital-acquired urinary tract infections [1]. Though less common than cystitis, APN is associated with significant morbidity and can lead to serious complications, including death. In the USA, it kills around 4000 people each year [2,3]. Evidence regarding the aetiology of this infection is mostly extrapolated from studies on cystitis [4]. European studies have reported an increase in antibiotic resistance in Gram-negative bacilli, especially *Escherichia coli* (*E. coli*), with frequent cross-resistance to fluoroquinolones, and ß-lactamase-producing strains [5,6]. The increasing presence of extended-spectrum ß-lactamase (ESBL)-producing *E. coli* strains in urine culture isolates of people with community-acquired APN is serious problem that leads to considerable use of healthcare resources [7,8,9]. Population ageing, increasing immunosuppression and the growing frequency of urinary catheterisation, among other factors, have given rise to multidrug-resistant microorganisms [10]. Previous studies have identified the following risk factors for developing ß-lactamase-producing strains: age over 55 years, prior use of antibiotics, prior urinary tract infections (UTIs), and diabetes mellitus [11,12]. Inadequate antibiotic therapy has been associated with increased morbidity [13,14]. Moreover, different studies have shown a wide variability in aetiology, depending on the place of acquisition, age, and comorbidities [2,15,16].

It is therefore crucial to regularly review APN-causing microorganisms and their sensitivity to antibiotics [17,18], and to identify the characteristics and factors associated with antimicrobial resistance [19] The literature contains very few studies on *E. coli* resistance in community-acquired APN in Spain or in the whole of Europe, and associated factors are rarely examined. In this study, we aimed to determine the prevalence of ESBL-producing *E. coli* in cases of community-acquired APN caused by *E. coli*, identify the factors associated with the presence of these strains and to use this information to design a explicative model for use in the determination of empirical antibiotic therapy regimens.

## 2. Materials and Methods

We conducted a cross-sectional study, analysing cases of community-acquired APN caused by *E. coli* that required hospital admission in Elda General University Hospital (Spain), which serves a population of 194,000 inhabitants (with 400 hospital beds, which has an infectious Disease Unit integrated into the internal medicine service, with 15 beds in its care). The study period spanned from 1 January 2012 to 31 June 2018. We included patients aged 14 and older in whom E. coli was isolated in urine or blood cultures. We excluded patients with no cultures, with negative results, in whom other microorganisms were isolated without *E. coli*, and who had incomplete information. We also excluded all cases of APN acquired in a care setting.

We searched for the APN diagnostic code in all electronic hospital discharge records created during the study period. After applying the inclusion criteria, we collected data related to demographic characteristics, comorbidities, Charlson comorbidity index, urinary pathology, urinary catheterisation, prior use of antibiotics, length of hospital stay, antimicrobial sensitivity, and prescribed empirical antibiotic therapy.

We applied the following definitions during data collection:

APN: a urinary tract infection infecting the upper urinary tract (renal pelvis and kidney parenchyma), usually causing fever, flank pain, nausea, vomiting, and clinical features of lower tract infection (frequent urination and, more rarely, tenesmus or incontinence).

Complicated APN: APN that worsens and leads to acute focal nephritis, renal corticomedullary abscess, perirenal abscess, papillary necrosis, or emphysematous pyelonephritis.

First admission: first time the patient was admitted with a primary diagnosis of APN.

ß-lactamase: an enzyme, produced by some bacteria, that confers resistance to ß-lactam antibiotics—such as penicillins, cephalosporins, monobactams and carbapenems (carbapenemases)—by hydrolysing the ß-lactam ring and generating a derivative without antimicrobial activity.

ESBLs: enzymes derived mainly from TEM and SHV-type enzymes (also described in CTX and OXA) and that can hydrolyse penicillins, broad-spectrum cephalosporins, and monobactams.

Positive urine culture result: >10^4^ colony-forming units (CFU) of *E. coli*.

Positive blood culture result: *E. coli* isolated in at least one blood culture.

Readmission: admission for the same reason within 30 days of discharge.

Relapse: recurrence of the disease after a period of remission or apparent recovery.

Sepsis: defined according to the 2012 Surviving Sepsis Campaign criteria [20], as the study period began in 2012.

Assuming a worst-case scenario of 50% prevalence, the sample size needed to estimate the proportion of ESBL-producing *E. coli* with a 95% confidence interval and a precision of 5% was 385 patients.

We performed a descriptive analysis by calculating absolute and relative frequencies for the categorical variables, and ranges, means and standard deviations for the quantitative variables. We analysed the factors associated with the presence of ESBL using contingency tables, applying a chi-squared test for the categorical variables, and comparing means with a Student’s *t*-test for the quantitative variables.

To quantify the association of each variable with the presence of ESBL, we fitted multivariate logistic models. Odds ratios (ORs) were calculated together with their 95% confidence intervals. Variables were selected in a stepwise procedure based on the Akaike Information Criterion. Goodness-of-fit and predictive performance were measured using the ROC curve.

For each cutoff value for the probability of presenting with ESBL, obtained through the multivariate logistic regression equation, we calculated validity indicators (sensitivity and specificity), predictive values and likelihood ratios. For all these indicators, we calculated 95% confidence intervals. To interpret the cutoff values and thus confirm or reject the diagnosis of APN caused by ESBL-producing *E. coli*, we applied evidence-based medicine criteria.

All analyses were performed with SPSS version 25 and R version 3.5.1.

## 3. Results

We reviewed 724 cases with a diagnosis of APN on discharge, of which 367 met the inclusion criteria of our study.

Figure 1 shows the reasons for excluding the remaining 357 patients.

Table 1 shows the baseline characteristics of the sample.

Most patients were women aged under 55 years (56.1%). One third of patients (33.5%) had at least one mild comorbidity, and 12% had taken antibiotics in the last three months.

Most patients (62.7%) were admitted to the short stay unit. Table 2 shows which departments patients were admitted to.

Most cases of APN were uncomplicated (90.1%). Eighteen patients (4.9%) developed sepsis and 99 (27%) had acute kidney injury. Blood cultures were performed in 247 patients (67.3%), and in 7.3% of all patients, ESBL-producing *E. coli* was isolated in blood cultures. The prevalence of ESBL-producing *E. coli* in cases of APN caused by *E. coli* was 13.9% (*n* = 51; 95% CI 10.4–17.4). Most cases were in women aged over 75 years, with statistically significant differences (Table 3). The prevalence of ESBL increases with age in female patients, reaching 26.9% in women over 75 years (Table 3).

Table 4 shows the multivariate logistic model for the presence of ESBL. Six variables (age, sex, smoking status, hypertension, urinary incontinence, recurrent UTIs) entered the model, giving a significant result (X2 66.4; *p* < 0.001). With the exception of age (*p* = 0.649) and urinary incontinence (*p* = 1.076), which acted as potential confounders, all variables showed statistical significance (*p* < 0.05), with ORs associated with increased likelihood of presence of ESBL. The variables with the highest odds radios were smoking status (OR = 4.846) and recurrent UTIs (OR = 4.673).

Figure 2 shows the ROC curve, with an area under the curve of 0.802 (95% CI 0.7307–0.8736).

The multivariate logistic regression equation was as follows:$$\mathrm{Probability}\;\mathrm{of}\;\mathrm{having}\;\mathrm{ESBL}=\frac{A}{1+A}$$
where A = exp [−3.7101 − 0.0049 AGE + 0.8309 SEX + 1.5781 SMOKING + 1.2066 HYPERTENSION + 0.8290 URINARY INCONTINENCE + 1.5417 RECURRENT UTIs], and where the items in the equation were defined as follows: age (years), sex (1 men, 0 woman), smoking (1 if smoker, 0 if nonsmoker or ex-smoker), hypertension (1 if yes, 0 if no), urinary incontinence (1 if yes, 0 if no), recurrent UTIs (1 if yes, 0 if no).

Table 5 and Table 6 show the different cutoffs for the probability of presenting with ESBL according to the model, with the following summary measures: sensitivity and specificity with 95% confidence intervals, Youden index, accuracy, positive and negative predictive values with 95% confidence intervals, and positive and negative likelihood ratios with 95% confidence intervals.

Facing a new patient, the multivariate logistic model shows the probability of the appearance of ESBL when replacing the values for the new patient with the variables in the multivariate model.

This probability takes values between 0 and 1. In order to generate a prediction about the possibility that the new patient will present with ESBL, a probability cutoff point is needed. Starting from that point, we will classify the new patient by ESBL presence.

A standard way to fix the cutoff point is to take 0.5, but also some predictive values, such as the sensibility and specificity, can be calculated for all the cutoff points between 0 and 1. In this way an optimal cut-off point can be chosen.

The optimal cutoff value corresponds to a 15% probability of having ESBL, with a sensitivity of 68.6% and a specificity of 78.7%. The positive likelihood ratios (LR+) that produce marked increases in the certainty of a positive ESBL-APN diagnosis (LR+ > 10) correspond to a probability of 40% or higher. The negative likelihood ratio (LR−) that produces moderate to marked increases in the certainty of a negative ESBL-APN diagnosis (LR− around 0.1) corresponds to a probability of 5% (Table 6).

## 4. Discussion

In our study, the prevalence of ESBL-producing *E. coli* in cases of community-acquired APN caused by *E. coli* was 13.9%, considering that we studied 367 APN of which 51 were ESBL. Male sex, smoking, hypertension, urinary incontinence, and recurrent UTIs were associated with the presence of ESBL-producing *E. coli*, and the model applied can correctly predict this outcome in 80.2% of cases.

Our prevalence is similar to the 12% prevalence observed by Talan et al. [21]. Data on APN are scarce: most publications provide global results from all isolates without specific reference to APN. In a recent study from Korea, ESBL-producing *E. coli* was isolated in up to 29% of cases of community-acquired APN [22,23]. In Spain, ESBL-producing strains of *E. coli* and *Klebsiella pneumoniae* have caused an increase in the prevalence of multidrug-resistant isolates of these bacteria in recent years, both in hospitals [10] and in the community [24]. In Europe, the prevalence of ß-lactamase-producing strains in community-acquired urinary tract infections is higher than in the USA, but lower that in Asia or South America [25].

Our study shows little fluctuation in prevalence since 2012, which is consistent with the steady 10% prevalence reported in another recent paper [11]. However, other studies have found an upward trend [25].

Most of our patients were women aged under 55 years; 42% had at least one comorbidity, and a high percentage were smokers. Almost one third had hypertension and one fifth had a history of urinary tract infections. Most patients were admitted to the short stay unit, corresponding to the normal length of a stay for uncomplicated APN. However, the patients admitted to the short stay unit had a lower percentage of ESBL-producing *E. coli* isolates than those admitted to the infectious disease unit. This shows that cases of greater complexity, in terms of clinical features and/or antibiotic therapy, tend to be admitted or transferred to specialised units.

The percentage of complicated APN in our patients (9.9%) was lower than in previous studies, possibly because these studies included patients with hospital-acquired as well as community-acquired APN and adopted a broader definition of complicated APN [21,26].

In the multivariate analysis, age was associated with the presence of ESBL-producing *E. coli*, as in previous studies [11]. A case–control study by Sun Hee Park et al. showed that age, prior use of antibiotics, diabetes and recurrent UTIs were independent risk factors for developing APN caused by ESBL-producing *E. coli*. In our analysis, this prevalence increased with patient age in women only, ranging from 8% in those aged under 55 years to 26% in those aged over 75 years. The higher number of comorbidities and greater exposure to antibiotics probably contribute to the higher prevalence of this resistant strain in older people [27,28,29,30]. Although this association did not show statistical significance in the multivariate analysis, we believe age could be a relevant factor to consider when proposing empirical antibiotic therapy.

As in previous studies [12,30,31,32], we found that patients with a history of UTIs were more likely to have ESBL-producing *E. coli*, which may be related to repeated use of antibiotics favouring the selection of multidrug-resistant microorganisms.

Although hypertension was prevalent in our sample, the multivariate analysis showed it to be an independent factor. We have found no other studies with similar results. Vascular damage caused by hypertension could lead to renal ischaemia and contribute to increasing susceptibility to infection, but this would not explain the appearance of resistance. Other factors associated with hypertension (e.g., older age, diabetes or prostate problems in men) could also play a role, although none of them showed statistical significance in the univariate analysis.

Smokers often have a wider range of comorbidities, including COPD, which can lead to respiratory infections. The antibiotics prescribed to treat these infections make patients more susceptible to antibiotic resistance [30].

The ROC curve of the multivariate model had a moderate area under the curve. The area under the curve is defined as the probability of correctly classifying a pair of randomly selected APN patients, one with ESBL-producing *E. coli* and one without, by applying the multivariate logistic regression equation. An area under the curve of 0.802 means that 80.2% of the time a randomly selected individual from the group of patients with APN caused by ESBL-producing *E. coli* will have a higher risk score estimated by the multivariate model than a randomly selected individual from the group with APN caused by non-ESBL *E. coli*.

The model obtained from the multivariate analysis together with the probability of having ESBL-producing *E. coli*, calculated using the logistic equation, and the summary measures for each probability cutoff value (sensitivity, specificity, predictive values and likelihood ratios) are highly relevant for clinical practice. The cutoff value for the probability of having ESBL-producing *E. coli* that maximises sensitivity and specificity is 0.15, but any other cutoff can be chosen depending on whether we are aiming for higher sensitivity (fewer false negatives) or higher specificity (fewer false positives).

In clinical practice, if a patient presents with suspected APN, the variables of the model can be measured and entered into the equation to obtain the probability of ESBL-producing *E. coli* infection. If we wish to classify this new patient as having or not having APN due to ESBL-producing *E. coli*, we can choose a probability cutoff value and use it to classify the patient according to whether their probability is higher or lower than the chosen cutoff. For each cutoff value, the clinician can establish all the epidemiological indicators.

Our study has some limitations. Firstly, it was conducted in a single hospital and the results should be corroborated before extrapolation to other contexts. As it was a retrospective study, some data may have been missing, although all the model predictors were present before the appearance of APN. Additionally, to ensure uniformity in data collection, we used the 2012 definition of sepsis, which is now considered outdated. Another limitation concerns the exclusion of APN patients who were not admitted to hospital, which may have resulted in the underreporting of cases. Unfortunately, we were unable to include these patients owing to limited availability of outpatient data.

Another limitation was that we did not have information on previous therapy received by each patient. We analysed colonisation by resistant microorganisms, but found no statistically significant association with ESBL-producing *E. coli* isolates in urine or blood cultures.

The main strength of our study is that it provides data on resistance in a specific infectious pathology, filling an information gap in Spain.

Our methodology is robust, and we built an explanatory model to help clinicians choose the best empirical antibiotic therapy.

In conclusion, the prevalence ESBL-producing *E. coli* in patients with APN caused by *E. coli* in our study was 13% and did not vary over the years. Male sex, smoking, hypertension, urinary incontinence, and recurrent UTIs were associated with the presence of ESBL-producing *E. coli* in people admitted to our hospital with community-acquired APN caused by *E. coli*. The multivariate model could be useful in clinical practice for diagnosing APN caused by ESBL-producing *E. coli* with moderate discriminative capacity.

## Figures and Tables

**Figure 1 jcm-10-05192-f001:**
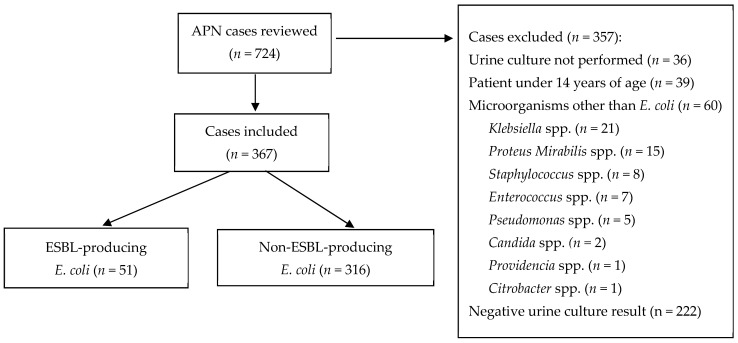
APN cases analysed, included, and excluded, with reasons for exclusion.

**Figure 2 jcm-10-05192-f002:**
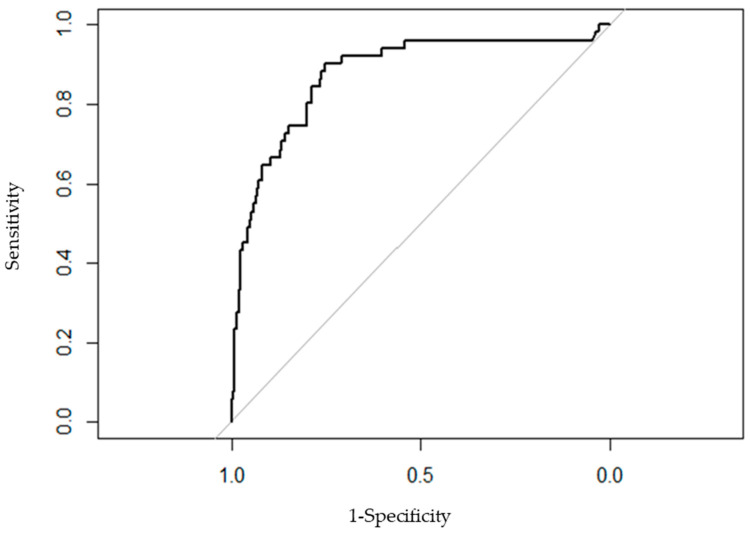
ROC curve of the adjusted multivariate model.

**Table 1 jcm-10-05192-t001:** Baseline characteristics of patients included in the study.

Variable	*n*	%
Year of admission		
2012	41	(11.2%)
2013	62	(16.9%)
2014	60	(16.3%)
2015	62	(16.9%)
2016	61	(16.6%)
2017	60	(16.3%)
2018 *	21	(5.7%)
Age		
<55 years	206	(56.1%)
55–74 years	93	(25.3%)
≤75 years	68	(18.5%)
Sex		
female	293	(79.8%)
male	74	(20.2%)
Type of admission		
First admission	359	(97.8%)
Readmission	8	(2.2%)
Smoker (yes)	151	(41.1%)
Alcohol (yes)	40	(10.9%)
General surgery (yes)	237	(64.6%)
CCI (age-adjusted)		
0	176	(48.0%)
1–2 (mild)	123	(33.5%)
≤3 (severe)	68	(18.5%)
Hypertension (yes)	131	(35.8%)
Diabetes mellitus (yes)	62	(16.9%)
Dependent ** (yes)	25	(6.8%)
Solid tumour (yes)	19	(5.2%)
Cardiovascular disease (yes)	53	(14.4%)
Chronic kidney disease (yes)	20	(5.4%)
Urinary incontinence (yes)	19	(5.2%)
Use of antibiotics in last 3 months (yes)	44	(12.0%)
Recurrent urinary tract infections (yes)	78	(21.3%)

* APN cases included from 1 January 2018 to 31 June 2018. ** Dependent (the patient need help for basic daily routines.

**Table 2 jcm-10-05192-t002:** Departments where APN patients were admitted.

Department	*n*	(%)
Short stay unit	230	(62.7)
Urology	36	(9.8)
Internal medicine	25	(6.8)
Emergency department	16	(4.4)
Other	14	(3.8)

**Table 3 jcm-10-05192-t003:** Prevalence of extended-spectrum ß-lactamase (ESBL)-producing *Escherichia coli* (*E. coli*) in patients with acute pyelonephritis caused by *E. coli*, by sex and age.

Scheme	Non-ESBL *E. coli*		ESBL *E. coli*		*p* Value
	*n*	%	*n*	%	
Women aged <55 years	163	91.6%	15	8.4%	<0.001
Women aged 55–74 years	56	88.9%	7	11.1%	
Women aged ≥75 years	38	73.1%	14	26.9%	
Men aged <55 years	25	89.3%	3	10.7%	0.186
Men aged 55–74 years	21	70.0%	9	30.0%	
Men aged ≥75 years	13	81.2%	3	18.8%	

**Table 4 jcm-10-05192-t004:** Multivariate logistic model for the presence of extended-spectrum ß-lactamases.

	Coefficient	OR	95% CI	*p* Value
Intercept	−3.7101			
Age	−0.0049	0.995	(0.974–1.017)	0.649
Sex (man)	0.8309	2.296	(1.043–5.055)	0.039
Smoker (yes)	1.5781	4.846	(2.376–9.882)	<0.001
Hypertension (yes)	1.2066	3.342	(1.423–7.852)	0.006
Urinary incontinence (yes)	0.8290	2.291	(0.689–7.618)	0.176
Recurrent UTIs (yes)	1.5417	4.673	(2.271–9.614)	<0.001

UTI: urinary tract infection.

**Table 5 jcm-10-05192-t005:** Sensitivity and specificity with 95% confidence intervals (CI), Youden index, and accuracy for the different cutoff values.

Cutoff	Sensitivity (95% CI)	Specificity (95% CI)	Youden Index	Accuracy
0.05	94.10 (87.6–100.6)	34.90 (29.6–40.2)	0.290	43.20
0.10	70.60 (58.1–83.1)	72.40 (67.5–77.3)	0.430	72.10
0.15	68.60 (55.9–81.3)	78.70 (74.2–83.2)	0.473	77.30
0.20	62.70 (49.4–76.0)	82.90 (78.7–87.1)	0.456	80.10
0.25	52.90 (39.2–66.6)	91.10 (88.0–94.2)	0.440	85.80
0.30	51.00 (37.3–64.7)	92.70 (89.8–95.6)	0.437	86.90
0.35	47.10 (33.4–60.8)	94.30 (91.7–96.9)	0.414	87.70
0.40	35.30 (22.2–48.4)	96.80 (94.9–98.7)	0.321	88.30
0.45	33.30 (20.4–46.2)	97.50 (95.8–99.2)	0.308	88.50
0.50	33.30 (20.4–46.2)	98.40 (97.0–99.8)	0.317	89.30
0.55	27.50 (15.2–39.8)	98.40 (97.0–99.8)	0.259	88.50
0.60	15.70 (5.7–25.7)	99.00 (97.9–100.1)	0.147	87.40
0.65	13.70 (4.3–23.1)	99.00 (97.9–100.1)	0.127	87.20
0.70	13.70 (4.3–23.1)	99.00 (97.9–100.1)	0.127	87.20

**Table 6 jcm-10-05192-t006:** Predictive values and likelihood ratios with 95% confidence intervals (CI).

Cutoff	PPV (95% CI)	NPV (95% CI)	LR+ (95% CI)	LR− (95% CI)
0.05	19.00 (14.2–23.8)	97.30(94.3–100.3)	1.45 (1.3–1.6)	0.17 (0.1–0.5)
0.10	29.30 (21.3–37.3)	93.80 (90.8–96.8)	2.56 (2.1–3.1)	0.41 (0.3–0.6)
0.15	34.30 (25.1–43.5)	93.90 (91.0–96.8)	3.22 (2.6–4.0)	0.40 (0.3–0.6)
0.20	37.20 (27.0–47.4)	93.20 (90.3–96.1)	3.67 (2.8–4.7)	0.45 (0.3–0.6)
0.25	49.10 (35.9–62.3)	92.30 (89.3–95.3)	5.94 (4.1–8.5)	0.52 (0.4–0.7)
0.30	53.10 (39.1–67.1)	92.10 (89.1–95.1)	6.99 (4.7–10.4)	0.53 (0.4–0.7)
0.35	57.10 (42.1–72.1)	91.70 (88.7–94.7)	8.26 (5.2–13.0)	0.56 (0.4–0.7)
0.40	64.30 (46.6–82.0)	90.20 (87.0–93.4)	11.03 (6.0–20.4)	0.67 (0.5–0.8)
0.45	68.00 (49.7–86.3)	90.00 (86.8–93.2)	13.32 (6.7–26.6)	0.68 (0.6–0.8)
0.50	77.30 (59.8–94.8)	90.10 (86.9–93.3)	20.81 (8.7–49.9)	0.68 (0.6–0.8)
0.55	73.70 (53.9–93.5)	89.30 (86.0–92.6)	17.19 (7.2–41.2)	0.74 (0.6–0.9)
0.60	72.70 (46.4–99.0)	87.90 (84.5–91.3)	15.70 (5.1–48.6)	0.85 (0.8–1.0)
0.65	70.00 (41.6–98.4)	87.60 (84.2–91.0)	13.70 (4.4–42.4)	0.87 (0.8–1.0)
0.70	70.00 (41.6–98.4)	87.60 (84.2–91.0)	13.70 (4.4–42.4)	0.87 (0.8–1.0)

PPV: positive predictive value; NPV: negative predictive value; LR+: positive likelihood ratio; LR−: negative likelihood ratio.

## Data Availability

All of the data used for this analysis can be confirmed at any time.

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
