# Peer review of "Factors Associated with Acute Community-Acquired Pyelonephritis Caused by Extended-Spectrum β-Lactamase-Producing Escherichia coli"

_jcm, 2021, doi:10.3390/jcm10215192_

Round 1
Reviewer 1 Report
Dear Authors,
you report a retrospective monocentric Spanish cohort of APN caused by ESBL-producing E. coli. Aim of the study was to determinate the prevalence of ESBL E. coli causing APN and to evaluate associated risk factors. As stated by the authors, the aim was to use data from the study in order to design a model able to aid clinicians in the choice of empirical antimicrobial therapy.
The study is interesting and well presented. Moreover, the study design and the statistical analysis used are solid.
However, my main concern regards the novelty of the study. In fact, risk factors associated with ESBL have already been explored in several studies, including larger multicentric cohorts. Furthermore, the aim of designing a model in order to help clinicians in the choice for empirical therapy was scarcely met, since data from the study do not add any relevant information with respect to previous studies. Moreover, it should be taken into account that clinical decision on the choice for empirical antibiotic therapy should be guided by national and local surveillance data on antimicrobial resistance alongside host characteristics. Thus, a clinical decision based exclusively on such model might not always be feasible and reliable.
Despite this relevant aspect, I believe that overall, the study presents some valuable points. In this regard, it could be considered for publication after some revisions.
Please find below some detailed comments.
GENERAL: Authors need to make some revisions regarding the correct use of English grammar, syntax, and punctuation.
TITLE: it should be specified that the study is on community-acquired APN
ABSTRACT: microorganisms should be written in italics and acronyms spelled for their first use.
INTRODUCTION:
L36: “In the USA, it kills around 4000 people each year”. This sentence in my opinion unnecessary. If you’d like to add some epidemiological data regarding APN, since the study was conducted in a European country local data might be of greater interest.
L45: “… and in some cases polymicrobial aetiology”. In my opinion unnecessary. Moreover, the way the sentence was formulated might make presume a polymicrobial aetiology being more dangerous than an infection by a multidrug-resistant organism.
L49: since you are focusing exclusively on E. coli infection, I would avoid this sentence on different aetiology
L53: While introducing the topic, before the aim, authors should state the gaps or need for this study.
MATERIALS AND METHODS:
L59: specify the country; specify hospital characteristics (how many beds, admission/year, presence of an Infectious Diseases Unit, availability of an Infectious Diseases consultation service)
L62: are polymicrobial infections with E.coli included in the study? If so, is it specified?
L66-69: did you collected data also on mortality, infectious diseases consultation received, directed antimicrobial therapy, prior hospitalisation, prior ESBL colonization? If so, please specify. Moreover, please clarify what “urinary conditions” stands for.
Other data collected not mentioned here: readmission, complicated vs. uncomplicated, relapse, concomitant sepsis
L82: 10^4
L86: I would add this definition right after the definition of APN.
RESULTS:
Fig.1: the list of microorganisms excluded should be homogeneous. Therefore, you should add the suffix spp. at all the microorganisms.
Table 1: sometimes authors add % and sometimes they do not. Please be consistent. Surgery, which kind of surgery? Presumably urological interventions solely or other kinds of surgery (e.g. abdominal)? Please specify.
Charlson Comorbidity Index: severe ³ 3; did you use the age-adjusted or age-unadjusted? Please specify
Table 1: I do not understand the variable “dependent”; please specify in the methods.
L155: not all the antibiotics select ESBLs in the same way. Therefore, knowing which class of antibiotics was used before hospitalization it would be of great interest. If you have these data please include in the manuscript.
L165: “7.3% of all patients gave a positive blood culture result”. Reword this sentence, in the current form does not flow well.
Table 2: it would be easier to read the table if departments were put into order of % of admission (from the highest % to the lowest) or alphabetically
Table3: 81.3% + 18.8% = 100.1% please review. Why didn’t you put all the p-values for all age classes? The third variable should not be >75 years?
Figure 2: ROC curve written in Spanish
Table 4: factors were adjusted for confounders? Why do you present only OR?
Table 5: necessary? Could not be included in the supplementary matherials?
L179: why Charlson score was not included in the model?
L213-216: In my opinion, in the current form, this part of results are not clearly presented. the statistical method behind this analysis (table 5 and 6) should be better explained in the methods section.
DISCUSSION:
L 259-262: this sentence should be rephrased since in the current form is not correct
L293-296: “Although hypertension was very prevalent in our sample, the multivariate analysis showed it to be an independent factor. We have found no other studies with similar results. Vascular damage caused by hypertension could lead to renal ischaemia and increase susceptibility to infection” In my opinion, this sentence should be rephrased. In fact, the correlation between hypertension and increased risk of ESBL APN could be due to factors other than the vascular damage as for instance the host characteristics. Therefore, I suggest using a more cautious sentence. Agreed
L299: better explain your hypothesis between smoking and APN caused by ESBL E. coli. Do you mean that smokers present a higher range of comorbidities and are therefore more prone to infections like APN?
L322-327: Limitations. In my opinion some important data are missing such as detailed previous antimicrobial therapy by classes and ESBL colonization. This has to be added in the limitation section. Please include a paragraph on the strengths of the study.
ETHICS APPROVAL: Please specify the number of protocol of Ethics committee approval of your institution.
Author Response
Dear editors and reviewers,
you report a retrospective monocentric Spanish cohort of APN caused by ESBL-producing E. coli. Aim of the study was to determinate the prevalence of ESBL E. coli causing APN and to evaluate associated risk factors. As stated by the authors, the aim was to use data from the study in order to design a model able to aid clinicians in the choice of empirical antimicrobial therapy.
The study is interesting and well presented. Moreover, the study design and the statistical analysis used are solid.
However, my main concern regards the novelty of the study. In fact, risk factors associated with ESBL have already been explored in several studies, including larger multicentric cohorts. Furthermore, the aim of designing a model in order to help clinicians in the choice for empirical therapy was scarcely met, since data from the study do not add any relevant information with respect to previous studies. Moreover, it should be taken into account that clinical decision on the choice for empirical antibiotic therapy should be guided by national and local surveillance data on antimicrobial resistance alongside host characteristics. Thus, a clinical decision based exclusively on such model might not always be feasible and reliable.
We partially agree with your comments regarding the novelty of our study; however we have not found similar studies on community-acquired APN in Europe or in Spain. Most studies report prevalence of resistance to all isolates and do not refer to a specific pathology.
We have not found any study published in the last 10 years in our field using scores or predictive models, which we believe makes our manuscript stand out.
The model is a tool for helping clinicians to take decisions, but could never replace their medical judgement, which should also take local epidemiologic data into account.
However, if we know the prevalence of causal micro-organism resistance in a given process, the choice of empiric antibiotic treatment will be better informed.
Our team conducted a similar study (pending publication) on the resistance of E. coli isolates in community-acquired APN.
Our guidelines are based on these results, but we considered the information too excessive to be included in the present study.
Clearly, clinical decision-making should never be based exclusively on the model: it is a very helpful tool, but should always be used in conjunction with the rest of the information.
Despite this relevant aspect, I believe that overall, the study presents some valuable points. In this regard, it could be considered for publication after some revisions.
Please find below some detailed comments.
GENERAL: Authors need to make some revisions regarding the correct use of English grammar, syntax, and punctuation. Amended.
TITLE: it should be specified that the study is on community-acquired APN. Amended.
ABSTRACT: microorganisms should be written in italics and acronyms spelled for their first use. Amended.
INTRODUCTION:
L36: “In the USA, it kills around 4000 people each year”. This sentence in my opinion unnecessary. If you’d like to add some epidemiological data regarding APN, since the study was conducted in a European country local data might be of greater interest.
We have not found any studies on community-acquired APN mortality in Europe or in Spain. There are studies in the paediatric population, but the rest report prevalence data, and in general refer to all urinary tract infections, with very few referring specifically to APN. Therefore, we believe that the reference to the USA can be maintained.
L45: “… and in some cases polymicrobial aetiology”. In my opinion unnecessary. Moreover, the way the sentence was formulated might make presume a polymicrobial aetiology being more dangerous than an infection by a multidrug-resistant organism.
Deleted as per your comment.
L49: since you are focusing exclusively on E. coli infection, I would avoid this sentence on different aetiology
In our opinion this sentence should not be deleted because it gives a general picture of PNA resistances.
L53: While introducing the topic, before the aim, authors should state the gaps or need for this study.
The literature contains very few studies on E. coli resistance in community-acquired APN in Spain or in the whole of Europe, and associated factors are rarely examined.
MATERIALS AND METHODS:
L59: specify the country; specify hospital characteristics (how many beds, admission/year, presence of an Infectious Diseases Unit, availability of an Infectious Diseases consultation service)
Amended.
L62: are polymicrobial infections with E.coli included in the study? If so, is it specified?
The polymicrobial infections have been excluded.
L66-69: did you collected data also on mortality, infectious diseases consultation received, directed antimicrobial therapy, prior hospitalisation, prior ESBL colonization? If so, please specify. Moreover, please clarify what “urinary conditions” stands for.
Other data collected not mentioned here: readmission, complicated vs. uncomplicated, relapse, concomitant sepsis
We included patients admitted to the Infectious Diseases Unit; the Internal Medicine Department and the Short Stay Unit work independently.
The Urology Department is frequently consulted about directed anti-microbial therapy once the multiresistant agent has been isolated, but the decision on empiric therapy for all cases of APN is taken in the Emergency Department.
We studied nasal and perineal colonization by resistant micro-organisms but found few positive results and no significant associations. We therefore decided not to include this information in our study. Nor did we include an examination of suitable and unsuitable empirical antibiotic therapy, as this topic is being covered in another study.
We have replaced the word “condition” with “pathology”, referring to any pathology of the urinary tract (e.g. urolithiasis, prostate problems). We collected some other data, but as they produced no relevant findings in the statistical analysis, we decided not to include them so as not to overcrowd the manuscript.
L82: 10^4 Amended.
L86: I would add this definition right after the definition of APN. Amended.
RESULTS:
Fig.1: the list of microorganisms excluded should be homogeneous. Therefore, you should add the suffix spp. at all the microorganisms. Amended
Table 1: sometimes authors add % and sometimes they do not. Please be consistent. Surgery, which kind of surgery? Presumably urological interventions solely or other kinds of surgery (e.g. abdominal)? Please specify. Amended
Charlson Comorbidity Index: severe ³ 3; did you use the age-adjusted or age-unadjusted? Please specify. Amended
Table 1: I do not understand the variable “dependent”; please specify in the methods.
Dependent: patients who required help for basic activities of daily living.
L155: not all the antibiotics select ESBLs in the same way. Therefore, knowing which class of antibiotics was used before hospitalization it would be of great interest. If you have these data please include in the manuscript.
This information is not available
L165: “7.3% of all patients gave a positive blood culture result”. Reword this sentence, in the current form does not flow well.
In 7.3 % of all patients, ESBL-producing E.coli was isolated in blood cultures.
Table 2: it would be easier to read the table if departments were put into order of % of admission (from the highest % to the lowest) or alphabetically
Amended
Table3: 81.3% + 18.8% = 100.1% please review. Why didn’t you put all the p-values for all age classes? The third variable should not be >75 years?
The value has been corrected from 81.3% to 81.2%. Thank you very much for spotting this rounding error.
Regarding p-values, Table 3 shows the association between age and having or not having ESBL. Age is a variable with 3 categories, and ESBL is a variable with 2 categories. The table corresponding to the crossover between both variables is a 3x2 table. The p-value is the result of a single hypothesis test, with the null hypothesis stating that the prevalences of ESBL are equal in all three age groups. This is why there is no p-value for each age group.
This crossover is performed separately for women and men, so there are 2 p-values.
The third age group is ≥75 years, as the previous group is 55-74 years.
Figure 2: ROC curve written in Spanish. Amended
Table 4: factors were adjusted for confounders? Why do you present only OR?
Yes, we adjusted for confounders.
Table 4 shows the multivariate logistic model with the significant predictors and also with the confounding factors we detected (age and urinary incontinence).
These factors are not significant, but as they influence the association between the predictors and the outcome, we left them in the model.
This table shows the coefficient of the logistic model, the ORs, and the 95% CIs and p-values for the ORs.
Table 5: necessary? Could not be included in the supplementary matherials?
In our opinion, table 5 helps the reader to better understand the results and conclusions. But if you think this table is not necessary, we can delete it.
L179: why Charlson score was not included in the model?
The Charlson Score was not included because the univariate analysis result was non-significant.
L213-216: In my opinion, in the current form, this part of results are not clearly presented. the statistical method behind this analysis (table 5 and 6) should be better explained in the methods section.
Additional information has been added.
For each new patient, we can calculate the probability of them carrying ESBL by inserting the corresponding values into the multivariate model.
A probability cut-off point between 0 and 1 is needed to establish whether this patient is a likely carrier.
One standard way to establish the cut-off point is simply to use 0.5. Alternatively, predictive values such as the sensibility and specificity can be calculated for all cut-off points between 0 and 1 to select an optimal point.
DISCUSSION:
L 259-262: this sentence should be rephrased since in the current form is not correct
Data on APN are scarce: most publications provide global results from all isolates without specific reference to APN. In a recent study from Korea, ESBL-producing E coli was isolated in up to 29% of cases of community-acquired APN
L293-296: “Although hypertension was very prevalent in our sample, the multivariate analysis showed it to be an independent factor. We have found no other studies with similar results. Vascular damage caused by hypertension could lead to renal ischaemia and increase susceptibility to infection” In my opinion, this sentence should be rephrased. In fact, the correlation between hypertension and increased risk of ESBL APN could be due to factors other than the vascular damage as for instance the host characteristics. Therefore, I suggest using a more cautious sentence. Agreed
Although hypertension was prevalent in our sample, the multivariate analysis showed it to be an independent factor. We have found no other studies with similar results. Vascular damage caused by hypertension could lead to renal ischaemia and contribute to increasing susceptibility to infection, but this would not explain the appearance of resistance. Other factors associated with hypertension (e.g. older age, diabetes, or prostate problems in men) could also play a role, although none of them showed statistical significance in the univariate analysis.
L299: better explain your hypothesis between smoking and APN caused by ESBL E. coli. Do you mean that smokers present a higher range of comorbidities and are therefore more prone to infections like APN?
Smokers often have a wider range of comorbidities, including COPD, which can lead to respiratory infections. The antibiotics prescribed to treat these infections make patients more susceptible to antibiotic resistance.
L322-327: Limitations. In my opinion some important data are missing such as detailed previous antimicrobial therapy by classes and ESBL colonization. This has to be added in the limitation section. Please include a paragraph on the strengths of the study.
Another limitation was that we did not have information on the previous therapy received by each patient. We analysed colonisation by resistant microorganisms, but found no statistically significant association with ESBL-producing E. coli isolates in urine or blood cultures.
The main strength of our study is that it provides data on resistance in a specific infectious pathology, filling an information gap in Spain.
Our methodology is robust and we built an explanatory model to help clinicians choose the best empirical antibiotic therapy.
ETHICS APPROVAL: Please specify the number of protocol of Ethics committee approval of your institution (2021/31EO).
Reviewer 2 Report
Major Comments.
- What diagnose codes in specific did you use to locate the APN diagnose?
- Please use the STROBE checklist.
- Please modify the conclusion in the abstract and the discussion section in regard to the study population of 367 APN of which 51 are ESBL.
Minor Comments
Please use E. coli instead of E. coli throughout the manuscript.
Author Response
Comments and Suggestions for Authors
Major Comments.
What diagnose codes in specific did you use to locate the APN diagnose?
We used diagnosis code CIE-10; N 10, N 11, N 11,1, N 11,8, N 11,9, N 12, N 13,2, N 16, N 20
Please use the STROBE checklist. Ok
Please modify the conclusion in the abstract and the discussion section in regard to the study population of 367 APN of which 51 are ESBL.
Amended
Minor Comments
Please use E. coli instead of E. coli throughout the manuscript.
Amended
STROBE Statement—checklist of items that should be included in reports of observational studies
|
|
Item No |
Recommendation |
|
Title and abstract |
1 |
(a) Indicate the study’s design with a commonly used term in the title or the abstract retrospective analysis of patients with community-acquired APN |
|
(b) Provide in the abstract an informative and balanced summary of what was done and what was found The prevalence of ESBL-producing E. coli was 13%. In the multivariate analysis, the factors in-dependently associated with ESBL were male sex (OR 2.296; 95% CI 1.043-5.055), smoking (OR 4.846, 95% CI 2.376-9.882), hypertension (OR 3.342, 95% CI 1.423-7.852), urinary incontinence (OR 2.291, 95% CI 0.689-7.618) and recurrent urinary tract infections (OR 4.673, 95% CI 2.271-9.614). The area under the ROC curve was 0.802 (IC 95% 0.7307 – 0.8736), meaning our model can cor-rectly classify an individual with ESBL-producing E. coli infection in 80.2% of cases. |
||
|
Introduction |
||
|
Background/rationale |
2 |
Explain the scientific background and rationale for the investigation being reported European studies have reported an increase in antibiotic resistance in gram-negative bacilli, especially Escherichia coli (E.coli), with frequent cross-resistance to fluoroquin-olones, and ß lactamase-producing strains [5, 6]. The increasing presence of extend-ed-spectrum ß-lactamase (ESBL)-producing E. coli strains in urine culture isolates of people with community-acquired APN is serious problem that leads to considerable use of healthcare resources [7, 8 ,9]. Population ageing, increasing immunosuppression and the growing frequency of urinary catheterisation, among other factors, have given rise to multidrug-resistant microorganisms [10]. Previous studies have identified the fol-lowing risk factors for developing ß-lactamase producing strains: age over 55 years, prior use of antibiotics, prior urinary tract infections (UTIs), and diabetes mellitus [11,12]. Inadequate antibiotic therapy has been associated with increased morbidity [13,14]. Moreover, different studies have shown a wide variability in aetiology, de-pending on the place of acquisition, age, and comorbidities [2,15,16]. |
|
Objectives |
3 |
State specific objectives, including any prespecified hypotheses In this study, we aimed to determine the prevalence of ESBL-producing E. coli in cases of community-acquired APN caused by E. coli and identify the factors associated with the presence of these strains; and to use this information to design a explicative model for use in the determination of empirical antibiotic therapy regimens |
|
Methods |
||
|
Study design |
4 |
Present key elements of study design early in the paper We conducted a cross-sectional study, analysing cases of community-acquired APN caused by E. coli that required hospital admission |
|
Setting |
5 |
Describe the setting, locations, and relevant dates, including periods of recruitment, exposure, follow-up, and data collection In Elda General University Hospital (Spain), which serves a population of 194 000 in-habitants (with 400 hospital beds, which has an infectious Disease Unit integrated into the internal medicine service, with 15 beds in its care). The study period spanned from January 1, 2012 to June 31, 2018. |
|
Participants |
6 |
(a) Cohort study—Give the eligibility criteria, and the sources and methods of selection of participants. Describe methods of follow-up Case-control study—Give the eligibility criteria, and the sources and methods of case ascertainment and control selection. Give the rationale for the choice of cases and controls Cross-sectional study—Give the eligibility criteria, and the sources and methods of selection of participants We included patients aged 14 and older in whom E. coli was isolated in urine or blood cultures. We excluded patients with no cultures, with negative results, in whom other micro-organisms were isolated without E. coli, and who had incomplete information. We also excluded all cases of APN acquired in a care setting. |
|
(b) Cohort study—For matched studies, give matching criteria and number of exposed and unexposed Case-control study—For matched studies, give matching criteria and the number of controls per case |
||
|
Variables |
7 |
Clearly define all outcomes, exposures, predictors, potential confounders, and effect modifiers. Give diagnostic criteria, if applicable We collected data related to demographic characteristics, comorbidities, Charlson comorbidity index, urinary pathology, urinary catheterisation, prior use of antibiotics, length of hospital stay, antimicrobial sensitivity, and prescribed empirical antibiotic therapy. |
|
Data sources/ measurement |
8* |
For each variable of interest, give sources of data and details of methods of assessment (measurement). Describe comparability of assessment methods if there is more than one group APN: a urinary tract infection infecting the upper urinary tract (renal pelvis and kidney parenchyma), usually causing fever, flank pain, nausea, vomiting, and clinical features of lower tract infection (frequent urination and, more rarely, tenesmus or in-continence). First admission: first time the patient was admitted with a primary diagnosis of APN. ß-lactamase: an enzyme, produced by some bacteria, that confers resistance to ß-lactam antibiotics – such as penicillins, cephalosporins, monobactams and car-bapenems (carbapenemases) – by hydrolysing the ß-lactam ring and generating a de-rivative without antimicrobial activity. ESBLs: enzymes derived mainly from TEM and SHV-type enzymes (also described in CTX and OXA) and that can hydrolyse penicillins, broad-spectrum cephalosporins, and monobactams. Positive urine culture result: > 104 colony-forming units (CFU) of E. coli. Positive blood culture result: E. coli isolated in at least one blood culture. Readmission: admission for the same reason within 30 days of discharge. Complicated APN: APN that worsens and leads to acute focal nephritis, renal cor-ticomedullary abscess, perirenal abscess, papillary necrosis, or emphysematous pyelo-nephritis. Relapse: recurrence of the disease after a period of remission or apparent recovery. Sepsis: defined according to the 2012 Surviving Sepsis Campaign criteria (20), as the study period began in 2012. |
|
Bias |
9 |
Describe any efforts to address potential sources of bias Bias are commented in discussion |
|
Study size |
10 |
Explain how the study size was arrived at Assuming a worst-case scenario of 50% prevalence, the sample size needed to es-timate the proportion of ESBL-producing E. coli with a 95% confidence interval and a precision of 5% was 385 patients. |
|
Quantitative variables |
11 |
Explain how quantitative variables were handled in the analyses. If applicable, describe which groupings were chosen and why We performed a descriptive analysis by calculating, means and standard deviations for the quantitative variables. Comparing means with the Student t test for the quantitative variables. |
|
Statistical methods |
12 |
(a) Describe all statistical methods, including those used to control for confounding To quantify the association of each variable with the presence of ESBL, we fitted multivariate logistic models. Odds ratios (ORs) were calculated together with their 95% confidence intervals. Variables were selected in a stepwise procedure based on the Akaike Information Criterion. Goodness-of-fit and predictive performance were measured using the ROC curve. For each cutoff value for the probability of having ESBL, obtained through the multivariate logistic regression equation, we calculated validity indicators (sensitivity and specificity), predictive values and likelihood ratios. For all these indicators, we calculated 95% confidence intervals All analyses were performed with SPSS version 25 and R version 3.5.1 |
|
(b) Describe any methods used to examine subgroups and interactions |
||
|
(c) Explain how missing data were addressed |
||
|
(d) Cohort study—If applicable, explain how loss to follow-up was addressed Case-control study—If applicable, explain how matching of cases and controls was addressed Cross-sectional study—If applicable, describe analytical methods taking account of sampling strategy |
||
|
(e) Describe any sensitivity analyses |
||
Continued on next page
|
Results |
||||||||||||||||||||||||
|
Participants |
13* |
(a) Report numbers of individuals at each stage of study—eg numbers potentially eligible, examined for eligibility, confirmed eligible, included in the study, completing follow-up, and analysed We reviewed 724 cases with a diagnosis of APN on discharge, of which 367 met the inclusion criteria of our study. We included 367 patients of which 51 are ESBL |
||||||||||||||||||||||
|
(b) Give reasons for non-participation at each stage Cases excluded (n=357): Urine culture not performed (n=36), Patient under 14 years of age (n=39), Microorganisms other than E. coli (n=60) |
||||||||||||||||||||||||
(c) Consider use of a flow diagram
|
||||||||||||||||||||||||
|
Descriptive data |
14* |
(a) Give characteristics of study participants (eg demographic, clinical, social) and information on exposures and potential confounders |
||||||||||||||||||||||
|
(b) Indicate number of participants with missing data for each variable of interest |
||||||||||||||||||||||||
|
(c) Cohort study—Summarise follow-up time (eg, average and total amount) |
||||||||||||||||||||||||
|
Outcome data |
15* |
Cohort study—Report numbers of outcome events or summary measures over time |
||||||||||||||||||||||
|
Case-control study—Report numbers in each exposure category, or summary measures of exposure |
||||||||||||||||||||||||
|
Cross-sectional study—Report numbers of outcome events or summary measures Most patients were women aged under 55 years (56.1%). One third of patients (33.5%) had at least one mild comorbidity, and 12% had taken antibiotics in the last three months. Most patients (62.7%) were admitted to the short stay unit Most cases of APN were uncomplicated (90.1%). Eighteen patients (4.9%) devel-oped sepsis and 99 (27%) had acute kidney injury. Blood cultures were performed in 247 patients (67.3%), and 7.3 % of whole studied patients was isolated the E.coli BLEE in the blood cultures. The prevalence of ESBL-producing E. coli in cases of APN caused by E.coli was 13.9% (n=51; 95% CI 10.4-17.4). Most cases were in women aged over 75 years, with statistically significant differences. The prevalence of ESBL increases with age in female patients, reaching 26.9% in women over 75 years. the multivariate logistic model for the presence of ESBL. Six variables (age, sex, smoking status, hypertension, urinary incontinence, recurrent UTIs) entered the model, giving a significant result (X2 66.4; p < 0.001). With the exception of age (p=0.649) and urinary incontinence (p=1.076), which acted as potential confounders, all variables showed sta-tistical significance (p<0.05), with ORs associated with increased likelihood of presence of ESBL. The variables with the highest odds radios were smoking status (OR=4.846) and recurrent UTIs (OR=4.673) The multivariate logistic regression equation was as follows: Probability of having ESBL=A/(1+A) where A = exp [ -3.7101 - 0.0049 AGE + 0.8309 SEX + 1.5781 SMOKING + 1.2066 HYPERTENSION + 0.8290 URINARY INCONTINENCE + 1.5417 RECURRENT UTIs] and where the items in the equation were defined as follows: age (years), sex (1 men, 0 woman), smoking (1 if smoker, 0 if non-smoker or ex-smoker), hypertension (1 if yes, 0 if no), urinary incontinence (1 if yes, 0 if no), recurrent UTIs (1 if yes, 0 if no). |
||||||||||||||||||||||||
|
Main results |
16 |
(a) Give unadjusted estimates and, if applicable, confounder-adjusted estimates and their precision (eg, 95% confidence interval). Make clear which confounders were adjusted for and why they were included The multivariate logistic model with the significant predictors and also with the confounding factors we detected (age and urinary incontinence). These factors are not significant, but as they influence the association between the predictors and the outcome, we left them in the model. This table shows the coefficient of the logistic model, the ORs, and the 95% CIs and p-values for the ORs. |
||||||||||||||||||||||
|
(b) Report category boundaries when continuous variables were categorized |
||||||||||||||||||||||||
|
(c) If relevant, consider translating estimates of relative risk into absolute risk for a meaningful time period |
||||||||||||||||||||||||
|
Other analyses |
17 |
Report other analyses done—eg analyses of subgroups and interactions, and sensitivity analyses |
||||||||||||||||||||||
|
Discussion |
||||||||||||||||||||||||
|
Key results |
18 |
Summarise key results with reference to study objectives In our study, the prevalence of ESBL-producing E. coli in cases of community-acquired APN caused by E. coli was 13.9%, considering that we studied 367 APN of which 51 were ESBL. Male sex, smoking, hypertension, urinary incontinence, and re-current UTIs were associated with the presence of ESBL-producing E. coli, and the model applied can correctly predict this outcome in 80.2% of cases. |
||||||||||||||||||||||
|
Limitations |
19 |
Discuss limitations of the study, taking into account sources of potential bias or imprecision. Discuss both direction and magnitude of any potential bias Our study has some limitations. Firstly, it was conducted in a single hospital and the results should be corroborated before extrapolation to other contexts. As it was a retrospective study, some data may have been missing, although all the model predic-tors were present before the appearance of APN. Additionally, to ensure uniformity in data collection, we used the 2012 definition of sepsis, which is now considered outdated. Another limitation concerns the exclusion of APN patients who were not admitted to hospital, which may have resulted in underreporting of cases. Unfortunately, we were unable to include these patients owing to limited availability of outpatient data. Another limitation was that we did not have information on the previous therapy received by each patient. We analysed colonisation by resistant microorganisms, but found no statistically significant association with ESBL-producing E. coli isolates in urine or blood cultures. The main strength of our study is that it provides data on resistance in a specific infectious pathology, filling an information gap in Spain. Our methodology is robust and we built an explanatory model to help clinicians choose the best empirical antibiotic therapy. |
||||||||||||||||||||||
|
Interpretation |
20 |
Give a cautious overall interpretation of results considering objectives, limitations, multiplicity of analyses, results from similar studies, and other relevant evidence |
||||||||||||||||||||||
|
Generalisability |
21 |
Discuss the generalisability (external validity) of the study results |
||||||||||||||||||||||
|
Other information |
||||||||||||||||||||||||
|
Funding |
22 |
Give the source of funding and the role of the funders for the present study and, if applicable, for the original study on which the present article is based |
||||||||||||||||||||||
*Give information separately for cases and controls in case-control studies and, if applicable, for exposed and unexposed groups in cohort and cross-sectional studies.
